# Potential impacts of climate change on agriculture and fisheries production in 72 tropical coastal communities

Joshua E. Cinner [1✉], Iain R. Caldwell [1], Lauric Thiault [2,3], John Ben[4], Julia L. Blanchard [5,6], Marta Coll [7], Amy Diedrich[8,9], Tyler D. Eddy [10], Jason D. Everett [11,12,13], Christian Folberth [14], Didier Gascuel [15], Jerome Guiet [16], Georgina G. Gurney[1], Ryan F. Heneghan [17], Jonas Jägermeyr[18,19,20], Narriman Jiddawi[21], Rachael Lahari[22], John Kuange[23], Wenfeng Liu [24], Olivier Maury [25], Christoph Müller [20], Camilla Novaglio [5,6], Juliano Palacios-Abrantes [26,27], Colleen M. Petrik [28], Ando Rabearisoa [29], Derek P. Tittensor[30,31], Andrew Wamukota[32] & Richard Pollnac[33,34]

Climate change is expected to profoundly affect key food production sectors, including fisheries and agriculture. However, the potential impacts of climate change on these sectors are rarely considered jointly, especially below national scales, which can mask substantial variability in how communities will be affected. Here, we combine socioeconomic surveys of 3,008 households and intersectoral multi-model simulation outputs to conduct a sub-national analysis of the potential impacts of climate change on fisheries and agriculture in 72 coastal communities across five Indo-Pacific countries (Indonesia, Madagascar, Papua New Guinea, Philippines, and Tanzania). Our study reveals three key findings: First, overall potential losses to fisheries are higher than potential losses to agriculture. Second, while most locations (> 2/3) will experience potential losses to both fisheries and agriculture simultaneously, climate change mitigation could reduce the proportion of places facing that double burden. Third, potential impacts are more likely in communities with lower socioeconomic status.

A full list of author affiliations appears at the end of the paper.

Climate change is expected to profoundly impact key food production sectors, with the tropics expected to suffer losses in both fisheries and agriculture. For example, by 2100 tropical areas could lose up to 200 suitable plant growing days per year due to climate change[1]. Likewise, fishable biomass in the ocean could drop by up to 40% in some tropical areas[2,3]. While understanding the magnitude of losses that climate change is expected to create in key food production sectors is crucial, it is the social dimensions of vulnerability that determine the degree to which societies are likely to be affected by these changes[4–8]. Vulnerability is the degree to which a system is susceptible to and unable to cope with the effects of change. It is comprised of exposure (the degree to which a system is stressed by environmental or social conditions), the social dimensions of sensitivity (the state of susceptibility to harm from perturbations), and adaptive capacity (people's ability to anticipate, respond to, and recover from the consequences of these changes)[4,9]. Together, the exposure and sensitivity domains are referred to as "potential impacts", which are the focus of this article.

Incorporating key social dimensions of vulnerability is particularly important because many coastal communities simultaneously rely on both agriculture and fisheries to varying degrees[10], yet assessments of climate change impacts and the policy prescriptions that come from them often consider these sectors in isolation[1,5,11–14]. Recently, studies have begun to look at the simultaneous impacts of climate change on both fisheries and agriculture at the national level[15,16], but this coarse scale does not capture whether people simultaneously engage with- and are likely to be affected by- changes in these sectors. Indeed, whether households engage in both fisheries and agriculture[10] will determine whether people have the knowledge, skills, and capital to substitute sectors if one declines, or alternatively, make them particularly susceptible to the potential double burden of a combined decline across sectors[15]. Thus, more localised analyses incorporating key social dimensions of vulnerability are required to better understand how combined impacts to fisheries and agriculture may affect coastal communities.

Here, we combine a measure of exposure based on model projections of losses to exploitable marine biomass (here dubbed fisheries catch potential) and agriculture from the Inter-Sectoral Impact Model Intercomparison Project (ISIMIP) Fast Track phase 3 dataset with a measure of sensitivity based on survey data about material wealth and engagement in fisheries, agriculture, and other occupational sectors from >3,000 households across 72 tropical coastal communities in five countries (see Supplementary Data file). We answer the following questions: 1) What are the potential impacts of projected changes to fisheries catch potential and agriculture on coastal communities?, 2) How much will mitigation measures reduce these potential impacts?, and 3) Are lower socioeconomic status coastal communities facing more potential impacts from climate change than their wealthier counterparts? We show that: fisheries tend to be more impacted than agriculture although there is substantial within-country variability; climate change mitigation can reduce the number of locations experiencing a double burden (i.e. losses to both fisheries and agriculture); and communities with lower socioeconomic status will experience the most severe climate change impacts.

## Results
Our study has three key results. First, we find that overall possible impacts on fisheries catch potential is higher than possible impacts on agriculture, but there can be substantial within-country variability in both exposure and sensitivity (Fig. 1). Specifically, exposure under the high-emissions Shared Socioeconomic Pathway 8.5 scenario (which has tracked historic

cumulative $CO_2$ emissions[17], but has been recently critiqued for over-projecting $CO_2$ emissions and economic growth[18]) indicates substantive losses by mid-century to fisheries catch potential [Fig. 1; 14.7% +/− 4.3% (SE) mean fisheries catch potential loss]. To put these projected losses in perspective, Sala et al[19]. found that strategically protecting 28% of the ocean could increase food provisioning by 5.9 million tonnes, which is just 6.9% of the 84.4 million tons of marine capture globally in 2018[20]. Thus, the mean expected fisheries catch potential losses are approximately double that which could be buffered by strategic conservation. Model run agreement about the directionality of change for projected impacts to fisheries catch potential was high (SSP5-8.5: 84.7 +/− 4.5% (SE); SSP1-2.6: 89.2 +/− 4.06% (SE)). Interestingly, crop models projected that agricultural productivity (based on rice, maize, and cassava- see methods) is expected to experience small average gains across the 72 sites (1.2% +/− 1.5% (SE) mean agricultural gain), with a large response range between sites and crops (Supplementary Fig. 1). However, the average gains are not significantly different from zero ($t = -0.80$, df = 5.0, $p = 0.46$), and model run agreement about directionality of change was lower for agriculture (SSP5-8.5: 69.1 +/− 4.82% (SE); SSP1-2.6: 70.4 +/− 3.27% (SE)). These projected agricultural gains are driven exclusively by rice (Supplementary Fig. 1), which has particularly large model disagreement[14,21]. Excluding rice shows an average decline in agricultural production by mid-century, since maize and cassava show consistent median losses under both SSP1-2.6 and SSP5-8.5 climate scenarios (Supplementary Fig. 1). Significantly greater losses in fisheries catch potential compared to agriculture productivity are apparent not only for our study sites (i.e. 15.9 +/− 5.6% (SE) greater; $t = 2.81$, df = 4.97, $p = 0.0379$), but also for a random selection of 4746 (10% of) coastal locations in our study countries with populations >25 people per $km^2$ (Fig. 2). Among those random sites, fisheries catch potential losses are an average of 15.6 +/− 5.1% (SE) greater than agriculture productivity changes ($t = 3.06$, df = 5.00, $p = 0.0282$). Differences between expected losses at our sites and the randomly selected sites are small for agriculture (Cohen's D for SSP5-8.5 = -0.31, SSP1-2.6 = −0.35) and negligible for fisheries catch potential (Cohen's D for SSP5-8.5 = -0.02, SSP1-2.6 = -0.03), indicating that our sites are not particularly biased towards high or low exposure for the study region. Not only is the level of exposure generally higher in fisheries compared to agriculture, but the sensitivity is on average nearly twice as high (Fig. 1a, b; 0.077 +/− 0.007 mean fisheries sensitivity; 0.04 +/− 0.01 mean agricultural sensitivity; $t = 3.0$, df = 2.26, $p$ value =0.0815).

Our analysis also reveals high within-country variability in potential impacts (i.e. both exposure and sensitivity), particularly for fisheries (Fig. 1) - a finding that may be masked in studies looking at national-level averages[15,16]. Looking only at the mean expected losses can obscure the more extreme fisheries catch potential losses projected for many communities (Figs. 1, 2). For example, under SSP5-8.5, our Indonesian sites are projected to experience very close to the average fisheries catch potential losses among our study sites (15.9 +/− 2.1%SE), but individual sites range from 6.5-32% losses (Fig. 1b). There is also substantial within-country variation in how communities are likely to experience climate change impacts, based on their sensitivity (Fig. 1a, b). For example, in the Philippines, exposure to fisheries is consistently moderate (range 8.9-12.6% loss), but sensitivity ranges from our lowest (0.001) to our highest recorded scores (0.32). There is also within-country variability in model agreement, particularly for the agricultural models in Indonesia, where agricultural model agreement ranges from 50-85% and fisheries model agreement ranges from 56-100% for SSP5-8.5, and 50-80% and 50-94%, respectively, for SSP1-2.6.

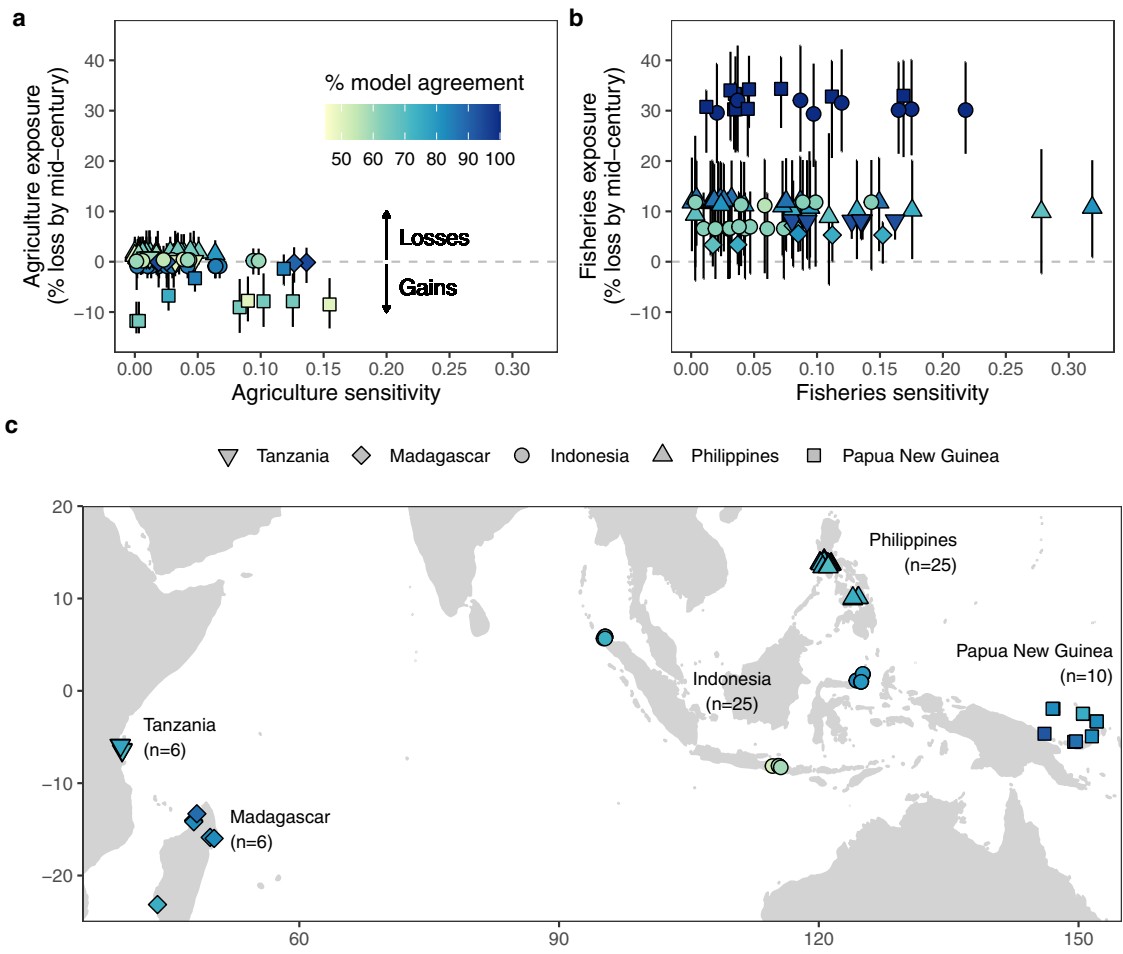

**Fig. 1 Potential impacts for (a) agriculture and (b) fisheries under SSP5-8.5 for (c) all study communities (n = 72).** Potential impacts comprise the exposure (y-axis, measured in potential losses, with error bars showing 25th and 75th percentiles) and sensitivity (x-axis, measured as level of dependence by households). Model run agreement (shown as colour gradient) highlights the proportion of (**a**) crop model runs (n = 20), (**b**) fisheries model runs (n = 16), and (**c**) average of agriculture and fisheries model runs that agree about the direction of change per site. Point shapes indicate country of each community. Inset map in Supplementary Fig. 9.

The second key result from our integrated assessment reveals that some locations will bear a double burden of losses to fisheries and agriculture simultaneously, but mitigation efforts that reduce greenhouse gas emissions could curb these losses. Specifically, under SSP5-8.5, 64% of our study sites are expected to lose productivity in fisheries and agriculture simultaneously (Fig. 3a), but this would reduce to 37% of sites under the low emissions scenario SSP1-2.6 (Fig. 3b). Again, the effect of mitigation is consistent in the random selection of 4746 sites (Supplementary Fig. 2), with 70% of randomly selected sites expected to experience a double burden under SSP5 8.5, and 47% under SSP1 2.6. Many of the sites expected to experience the highest losses to both fisheries catch potential and agriculture have moderate to high sensitivity (Fig. 3a, Supplementary Fig. 3), which means the impacts of these changes could be profoundly felt by coastal communities.

Over a third of our sites (36% under SSP5-8.5) are expected to experience increases in agriculture (due to $CO_2$ fertilization effects that fuel potential increases particularly in rice yields) while experiencing losses in fisheries catch potential. For these sites, a question of critical concern is whether the potential gains in agriculture could help offset the losses in fisheries catch potential. The answer to this lies in part in the degree of substitutability between sectors. Our survey of 3,008 households

reveals high variation among countries, and even within some countries in the degree of household occupational multiplicity incorporating both agriculture and fisheries sectors (Table 1). 31% of households in our study engaged in both fishing and agriculture, though this ranged from 10% of households in the Philippines to 77% of households in Papua New Guinea. This means that the degree to which agricultural gains might possibly offset some fisheries losses at the household scale is very context dependent. Our survey also revealed that 17% of households were involved in agriculture but not fisheries, ranging from 33% in Madagascar to 3% in our Papua New Guinean study communities. Alternatively, more than a third of households surveyed in Indonesia and Philippines were involved in fisheries but not agriculture (36% and 37% respectively), compared to a low value of 16% in Madagascar. In 12% of the Philippines communities surveyed (n = 3), not a single household was engaged in agriculture. Thus, for 32% of households across our sample, including some entire communities, potential agricultural gains will not offset potential fisheries losses. In these locations building adaptive capacity to buffer change will be critical[9].

Our third key result is that coastal communities with lower socioeconomic status are more likely to experience potential impacts than communities of higher socioeconomic status across the climate mitigation scenarios (SSP1-2.6 and SSP5-8.5; Fig. 4).

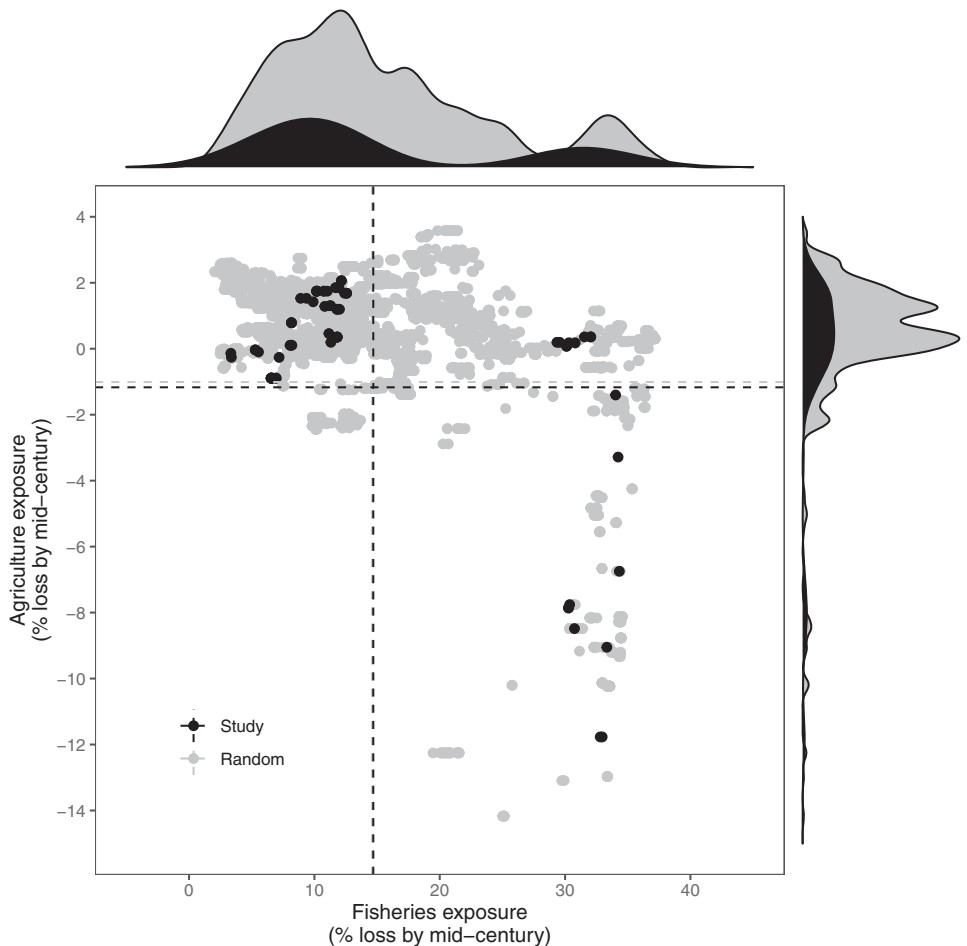

**Fig. 2 A comparison of expected fisheries catch potential and agriculture losses (exposure) by mid-century under SSP5-8.5.** Black dots, histograms, and dotted lines (for mean exposures) represent our study sites ($n = 72$). Grey dots, histograms, and dotted lines represent a random selection of 10% of coastal cells with population densities >25 people/km$^2$ from our study countries ($n = 4746$).

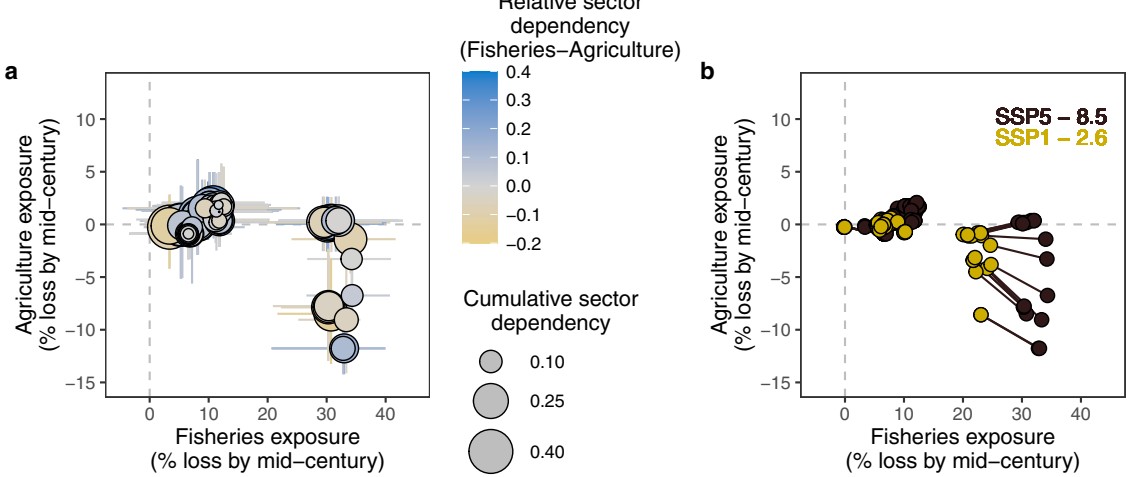

**Fig. 3 Simultaneous potential losses to fisheries and agriculture in coastal communities ($n = 72$). a** Under SSP5-8.5 agricultural losses (y-axis) plotted against fisheries losses (x-axis), with bubble size revealing the overall sensitivity and colour revealing the fisheries-agricultural relative sector dependency of each community's sensitivity. **b** Potential benefits of mitigation shown by the potential losses for each community change going from the high emissions scenario (SSP5-8.5 in red) to a low emissions scenario (SSP1-2.6 in yellow).

**Table 1 Proportion of surveyed households in each study country engaged in both agriculture and fisheries, agriculture but not fisheries, and fisheries but not agriculture.**

| Country | Number of Households | Agriculture and Fisheries | Agriculture, No Fisheries | Fisheries, No Agriculture |
|---|---|---|---|---|
| Indonesia | 1140 | 0.25 | 0.18 | 0.36 |
| Madagascar | 339 | 0.42 | 0.33 | 0.16 |
| Papua New Guinea | 318 | 0.77 | 0.03 | 0.18 |
| Philippines | 973 | 0.11 | 0.18 | 0.37 |
| Tanzania | 238 | 0.69 | 0.04 | 0.26 |

Note: proportions do not add up to 1 because some households were not engaged in agriculture or fisheries.

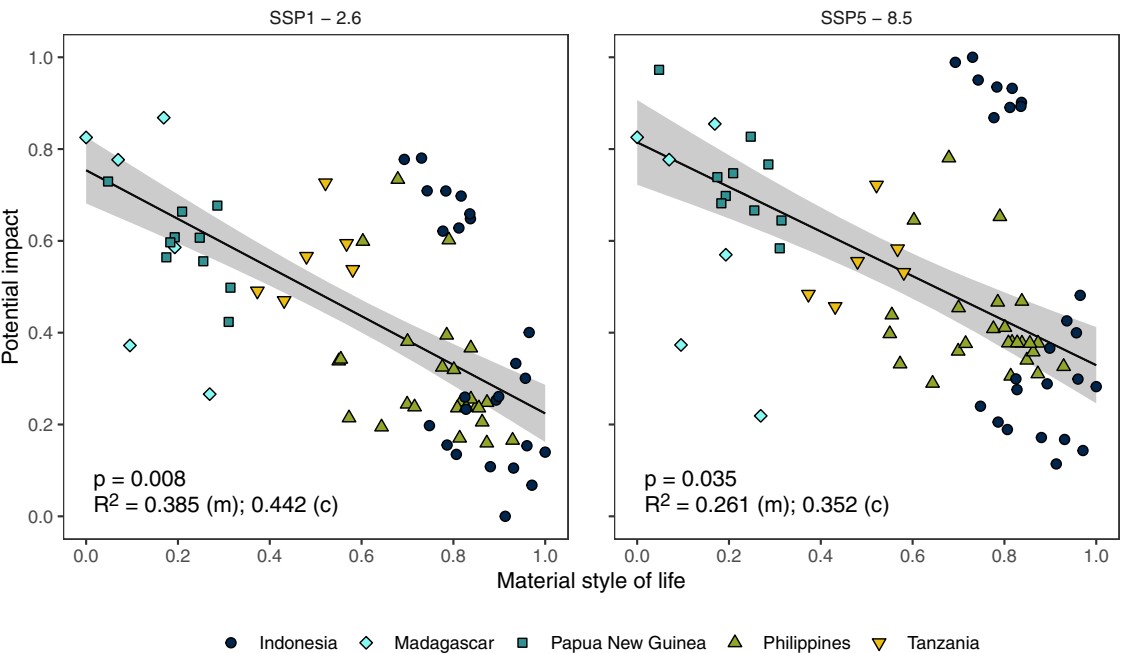

**Fig. 4 Relationships between potential impacts (calculated as the Euclidean distance of exposure and sensitivity) and material style of life (a metric of wealth based on material assets) under different mitigation strategies across all studied communities ($n = 72$).** Black lines are predictions from linear mixed-effects models (with country as random effect) and grey bands are standard errors. Statistical significance ($p$) and fit ($R^2$) of the mixed-effects models are also shown: (m) = marginal $R^2$, (c) = conditional $R^2$. Point shape and colour indicate country.

Specifically, we examined the relationship between the average material style of life (a metric of wealth based on material assets; see methods) in a community and the relative potential impacts of simultaneous fisheries catch potential and agriculture losses (measured as the Euclidean distance of sensitivity and exposure from the origin). Importantly, socioeconomic status is related to both sensitivity and exposure (Supplementary Fig. 4). In other words, low socioeconomic status communities tend to have higher sensitivity to fisheries and agriculture than the wealthy, and are significantly more likely to be exposed to climate change impacts. Our findings regarding the relationship between socioeconomic status and sensitivity are consistent with a broad body of literature that shows how people tend to move away from natural resource-dependent occupations as they become wealthier[10,22–25]. One potential interpretation of our findings is that alternative livelihood programs (e.g. jobs outside the fisheries or agricultural sectors, such as the service industry) could reduce sensitivity in lower socioeconomic status communities. However, decades of research on livelihood diversification has highlighted a multitude of reasons why alternative livelihood projects frequently fail[26], including that they do not provide high levels of non-economic satisfactions (e.g., social, psychological, and cultural)[27–29], as well as cultural barriers to switching occupations (e.g. caste systems)[30], and attachment to identity and place[31].

Alternative occupations need to provide some of the same satisfactions, including basic needs (safety, income), social and psychological needs (time away from home, community in which you live, etc.), and self-actualization (adventure, challenge, opportunity to be own boss, etc.). For example, fishing attracts individuals manifesting a personality configuration referred to as an externalizing disposition, which is characterized by a need for challenges, adventure, and risk. Fishing can be extremely satisfying for people with this personality complex, while many alternative occupations can lead to job dissatisfaction, which has negative social and psychological consequences[32,33]. Research has shown that recreational fishing captain or guide jobs produce some of the same satisfactions as fishing and have been successfully introduced as alternative occupations[33]. Despite these limited successes, alternative livelihood programs frequently fail and are not a viable substitute for mitigating climate change for the ~6 million coral reef fishers globally[34].

## Discussion
Our study is an important first step in examining the potential simultaneous impacts to fisheries catch potential and agriculture in coastal communities, but has some limitations, some of which could be addressed in future studies. First, our measure of

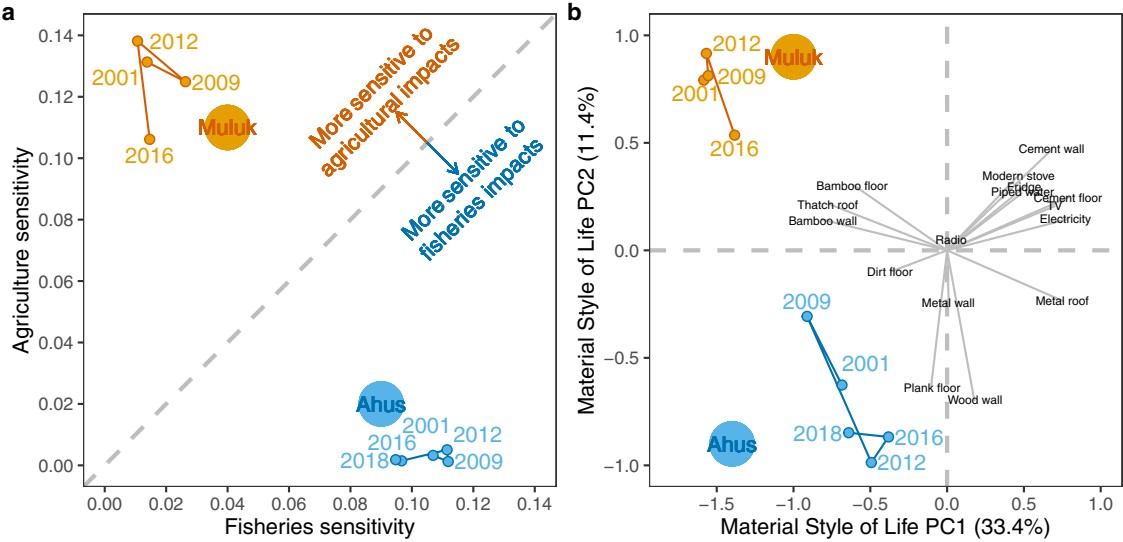

**Fig. 5 Changes in (a) agriculture-fisheries sensitivity and (b) material wealth over time in two Papua New Guinean communities: Muluk (orange) and Ahus (blue). b** shows how the communities change along the first two axes of a principal component analysis (i.e., PC1 and PC2), based on 16 household-scale material items, with black text and grey lines indicate the relative contribution of each material item to principal components.

exposure was dynamic (i.e., it was projected into the future), while our measures of sensitivity and material wealth were static (i.e., from a single point in time) and did not consider potential changes over time. Although there are projections of how national-scale measures of wealth (e.g. gross domestic product; GDP) may change in the future, there are no reliable projections for household- or community-scale changes to material wealth or livelihoods. As an additional analysis, we examined observed changes in sensitivity and material wealth over 15 and 16 years, respectively, in two Papua New Guinean coastal communities (Fig. 5). We found that, over the observed time frame (2001-2016), which is approximately half that of the predicted time frame of exposure, sensitivity scores were extremely stable, particularly in Ahus (Fig. 5). Similarly, material wealth was also reasonably stable over time, but did reflect a shift in both communities toward more houses being built out of sturdier material (e.g., wood plank walls and floor, metal roofs). Importantly, while there were absolute changes to material wealth in both communities, the relative position stayed very similar. Although these data do not allow us to make inferences about what will happen into the future, they do highlight that, at least in decadal timeframes, these indicators are reasonably stable. One alternative approach may have been to assume that projected national-scale changes to GDP would apply evenly across each coastal community within a country (i.e., adjust the intercept of both material wealth and correlated sensitivity for each country relative to the projected changes in GDP). However, given the wide spread of material wealth and sensitivity scores within countries, we ultimately were less comfortable with the assumptions inherent in the approach (i.e., that national-scale changes would affect all communities in a country equally) than with the caveat that our metrics were static.

Second, there are key limitations and assumptions to the models we used. For example, many tropical small-scale fisheries target seagrass[35] and coral reef habitats[34], which are not represented in the global ensemble models. Additionally, the ensemble models were developed at relatively low spatial resolution (e.g. 1° cells), and are not designed to capture higher-resolution structures and processes. Our approach for dealing with this was to make transparent the degree of ensemble model run agreement about the direction of change, which relies on the assumption that

we have greater confidence in projections that have higher model run agreement. Another limitation is that there may be discrepancies between the total consumer biomass (see method) in the absence of fishing that is outputed by the models used here and what would actually be harvested by fishers since total consumer biomass can include both target and non-target species as well as other taxa entirely. Despite these limitations, we assumed that total consumer biomass is directly related to potential fisheries yields[11]. Likewise, we included just three crops in the agricultural models (rice, maize, and cassava), which are key in the study region, with many study countries growing 2 or more of these crops. For example, in 2020 Indonesia was the 4th largest producer of rice in the world, the 5th largest producer of cassava, and the 8th largest producer of maize[36]. However, subsistence agriculture in Papua New Guinea is dominated by banana and yams, for which agricultural yield projections were not available. We used an unweighted average of projected changes in these three crops to represent a portfolio of small-scale agriculture, with a sensitivity test based on agricultural projections weighted by current yields/production area proportions of current yields (Supplementary Fig. 1). Finally, it is important to keep key model assumptions in mind when interpreting these data. For example, the agricultural models assumed no changes in farm management or climate change adaptation over time, while the fisheries models do not explicitly resolve predation impacts from higher trophic levels on phytoplankton.

Third, our sensitivity metric examined a somewhat narrow aspect of what makes people sensitive to climate change. Sensitivity is thought to contain dimensions of economic, demographic, psychological, and cultural dependency[37]. Our metric was based on people's engagement in natural resource-based livelihoods, which primarily captures the economic dimensions (although livelihoods do provide cultural and psychological contributions to people[26,28,29,31,38]).

Fourth, our study explicitly focused on the potential impacts of climate change in 72 Indo-Pacific coastal communities by examining their sensitivity and exposure, but our methodology did not enable us to incorporate adaptive capacity. Adaptive capacity is a latent trait that enables people to adapt to and take advantage of the opportunities created by change[39,40], and is critically important in determining the fate of coastal

communities under climate change. Adaptive capacity is thought to consist of dimensions of assets, flexibility, social organisation, learning, socio-cognitive, and agency[9,41,42]. Unfortunately, indicators of these dimensions of adaptive capacity were not collected in a standardised manner across all of the different projects comprising this study.

Fifth, we investigated the potential impacts of climate change on two key food production sectors, but there may be other climate change impacts which have much more profound impacts on people's wellbeing. For example, sea level rise may destroy homes and other infrastructure[43], while heat waves may result in direct mortality[44]. Last, we used shared socioeconomic pathway exploratory scenarios that bracket the full range of scenario variability (SSP5-8.5 and SSP1-2.6). At the time of publication, these were the only scenarios available for both fisheries and agriculture using the ISIMIP Fastrack Phase 3 dataset. Future publications may wish to explore additional scenarios.

Our study quantifies the potential impacts of climate change on key food production sectors in tropical coastal communities across a broad swath of the Indo-Pacific. We find that both exposure and sensitivity to fisheries is generally higher than to agriculture, but some places may experience losses from both sectors simultaneously. These losses may be compounded by other drivers of change, such as overfishing or soil erosion, which is already leading to declining yields[45,46]. Simultaneous losses to both fisheries catch potential and agriculture will limit people's opportunity to adapt to changes through switching livelihoods between food production sectors[9]. This will especially be the case in lower socioeconomic status communities where dependence on natural resources is higher[10]. Together, our integration of model projections and socioeconomic surveys highlight the importance of assessing climate change impacts across sectors, but reveals important mismatches between the scale at which people will experience the impacts of climate change and the scale at which modelled projections about climate change impacts are currently available.

## Methods

**Sampling of coastal communities**. Here, we integrated data from five different projects that had surveyed coastal communities across five countries[47–50]. Between 2009 and 2015, we conducted socioeconomic surveys in 72 sites from Indonesia ($n = 25$), Madagascar ($n = 6$), Papua New Guinea ($n = 10$), the Philippines ($n = 25$), and Tanzania (Zanzibar) ($n = 6$). Site selection was for broadly similar purposes- to evaluate the effects of various coastal resource management initiatives (collaborative management, integrated conservation and development projects, recreational fishing projects) on people's livelihoods in rural and peri-urban villages. Within each project, sites were purposively selected to be representative of the broad range of socioeconomic conditions (e.g., population size, levels of development, integration to markets) experienced within the region. We did not survey strictly urban locations (i.e., major cities). Because our sampling was not strictly random, care should be taken when attempting to make inferences beyond our specific study sites.

We surveyed between 13 and 150 households per site, depending on the population of the communities and the available time to conduct interviews per site. All projects employed a comparable sampling design: households were either systematically (e.g., every third house), randomly sampled, or in the case of three villages, every household was surveyed (a census) (see Supplementary Data file). Respondents were generally the household head, but could have been other household members if the household head was not available during the study period (i.e. was away). In the Philippines, sampling protocol meant that each village had an even number of male and female respondents. Respondents gave verbal consent to be interviewed.

The following standard methodology was employed to assess material style of life, a metric of material assets-based wealth[48,51]. Interviewers recorded the presence or absence of 16 material items in the household (e.g., electricity, type of walls, type of ceiling, type of floor). We used a Principal Component Analysis on these items and kept the first axis (which explained 34.2% of the variance) as a material wealth score. Thus, each community received a mean material style of life score, based on the degree to which surveyed households had these material items, which we then scaled from 0 to 1. We also conducted an exploratory analysis of how material style of life has changed in two sites in Papua New Guinea (Muluk and Ahus) over fifteen and sixteen-year time span across four and five-time

periods (2001, 2009, 2012, 2016, and 2002, 2009, 2012, 2016, 2018), respectively, that have been surveyed since 2001/2002[52]. These surveys were semi-panel data (i.e. the community was surveyed repeatedly, but we did not track individuals over each sampling interval) and sometimes occurred in different seasons. For illustrative purposes, we plotted how these villages changed over time along the first two principal components.

**Sensitivity**. We asked each respondent to list all livelihood activities that bring in food or income to the household and rank them in order of importance. Occupations were grouped into the following categories: farming, cash crop, fishing, mariculture, gleaning, fish trading, salaried employment, informal, tourism, and other. We considered fishing, mariculture, gleaning, fish trading together as the 'fisheries' sector, farming and cash crop as the 'agriculture' sector and all other categories into an 'off-sector'.

We then developed three distinct metrics of sensitivity based on the level of dependence on agriculture, fisheries, and both sectors together. Each metric incorporates the proportion of households engaged in a given sector (e.g., fisheries), whether these households also engage in occupations outside of this sector (agriculture and salaried/formal employment; referred to as 'linkages' between sectors), and the directionality of these linkages (e.g., whether respondents ranked fisheries as more important than other agriculture and salaried/formal employment) (Eqs. 1–3)

$$S_A = \frac{A}{A + NA} \times \frac{N}{A + NA} \times \frac{\left(\frac{r_a}{2}\right) + 1}{r_a + r_{na} + 1} \quad (1)$$

$$S_F = \frac{F}{F + NF} \times \frac{N}{F + NF} \times \frac{\left(\frac{r_f}{2}\right) + 1}{r_f + r_{nf} + 1} \quad (2)$$

$$S_{AF} = \frac{AF}{AF + NAF} \times \frac{N}{AF + NAF} \times \frac{\left(\frac{r_{af}}{2}\right) + 1}{r_{af} + r_{naf} + 1} \quad (3)$$

where $S_A$, $S_F$ and $S_{AF}$ are a community's sensitivity in the context of agriculture, fisheries and both sectors, respectively. A, F and AF are the number of households relying on agriculture-related occupations within that community, fishery-related and agriculture- and fisheries-related occupations within the community, respectively. NA, NF and NAF are the number of households relying on non-agriculture-related, non-fisheries-related, and non-agriculture-or-fisheries-related occupations within the community, respectively. N is the number of households within the community. $r_a$, $r_f$ and $r_{af}$ are the number of times agriculture-related, fisheries-related and agriculture-and-fisheries-related occupations were ranked higher than their counterpart, respectively. $r_{na}$, $r_{nf}$ and $r_{naf}$ are the number of times non-agriculture, non-fisheries, and non-agriculture-and-fisheries-related occupations were ranked higher than their counterparts. As with the material style of life, we also conducted an exploratory analysis of how joint agriculture-fisheries sensitivity has changed over time in a subset of sites (Muluk and Ahus villages in Papua New Guinea) that have been sampled since 2001/2002[52]. Although our survey methodology has the potential for bias (e.g. people might provide different rankings based on the season, or there might be gendered differences in how people rank the importance of different occupations[53]), our time-series analysis suggest that seasonal and potential respondent variation do not dramatically alter our community-scale sensitivity metric.

**Exposure**. To evaluate the exposure of communities to the impact of future climates on their agriculture and fisheries sectors, we used projections of production potential from the Inter-Sectoral Impact Model Intercomparison Project (ISIMIP) Fast Track phase 3 experiment dataset of global simulations. Production potential of agriculture and fisheries for each of the 72 community sites and 4746 randomly selected sites from our study countries with coastal populations >25 people/km² were projected to the mid-century (2046–2056) under two emission scenarios (SSP1-2.6, and SSP5-8.5) and compared with values from a reference historical period (1983–2013).

For fisheries exposure ($E_F$), we considered relative change in simulated total consumer biomass (all modelled vertebrates and invertebrates with a trophic level >1). For each site, the twenty nearest ocean grid cells were determined using the Haversine formula (Supplementary Fig. 5). We selected twenty grid cells after a sensitivity analysis to determine changes in model agreement based on different numbers of cells used (1, 3, 5, 10, 20, 50, 100; Supplementary Figs. 6–7), which we balanced off with the degree to which larger numbers of cells would reduce the inter-site variability (Supplementary Fig. 8). We also report 25th and 75th percentiles for the change in marine animal biomass across the model ensemble. Projections of the change in total consumer biomass for the 72 sites were extracted from simulations conducted by the Fisheries and marine ecosystem Model Intercomparison Project (FishMIP)[3,54]. FishMIP simulations were conducted under historical, SSP1-2.6 (low emissions) and SSP5-8.5 (high emissions) scenarios forced by two Earth System Models from the most recent generation of the Coupled Model Intercomparison project (CMIP6);[55] GFDL-ESM4[56] and IPSL-CM6A-LR[57]. The historical scenario spanned 1950–2014, and the SSP scenarios spanned 2015–2100. Nine FishMIP models provided simulations: APECOSM[58,59], BOATS[60,61], DBEM[2,62], DBPM[63], EcoOcean[64,65], EcoTroph[66,67], FEISTY[68],

Macroecological[69], and ZooMSS[11]. Simulations using only IPSL-CM6A-LR were available for APECOSM and DBPM, while the remaining 7 FishMIP models used both Earth System Model forcings. This resulted in 16 potential model runs for our examination of model agreement, albeit with some of these runs being the same model forced with two different ESMs. Thus, the range of model agreement could range from 8 (half model runs indicating one direction of change, and half indicating the other) to 16 (all models agree in direction of change). Model outputs were saved with a standardised 1° spatial grid, at either a monthly or annual temporal resolution.

For agriculture exposure ($E_A$), we used crop model projections from the Global Gridded Crop model Intercomparison Project (GGCMI) Phase 3[14], which also represents the agriculture sector in ISIMIP. We used a window of 11×11 cells centred on the site and removed non-land cells (Supplementary Fig. 5). The crop models use climate inputs from 5 CMIP6 ESMs (GFDL-ESM4, IPSL-CM6A-LR, MPI-ESM1-2-HR, MRI-ESM2-0, and UKESM1-0-LL), downscaled and bias-adjusted by ISIMIP and use the same simulation time periods. We considered relative yield change in three rain-fed and locally relevant crops: rice, maize, and cassava, using outputs from 4 global crop models (EPIC-IIASA, LPJmL, pDSSAT, and PEPIC), run at 0.5° resolution. These 4 models with 5 forcings generate 20 potential model runs for our examination of model agreement. Yield simulations for cassava were only available from the LPJmL crop model. All crop model simulations assumed no adaptation in growing season and fertilizer input remained at current levels. Details on model inputs, climate data, and simulation protocol are provided in ref. [14]. At each site, and for each crop, we calculated the average change (%) between projected vs. historical yield within 11×11 cell window. We then averaged changes in rice, maize and cassava to obtain a single metric of agriculture exposure ($E_A$).

We also obtained a composite metric of exposure ($E_{AF}$) by calculating each community's average change in both agriculture and fisheries:

$$E_{AF} = \frac{E_A + E_F}{2} \qquad (4)$$

**Potential Impact**. We calculated relative potential impact as the Euclidian distance from the origin (0) of sensitivity and exposure.

**Sensitivity test**. To determine whether our sites displayed a particular exposure bias, we compared the distributions of our sites and 4746 sites that were randomly selected from 47,460 grid cells within 1 km of the coast of the 5 countries we studied which had population densities >25 people/km$^2$, based on the SEDAC gridded populating density of the world dataset (https://sedac.ciesin.columbia.edu/data/set/gpw-v4-population-density-rev11/data-download).

We used Cohen's D to determine the size of the difference between our sites and the randomly selected sites.

**Validating ensemble models**. We attempted a two-stage validation of the ensemble model projections. First, we reviewed the literature on downscaling of ensemble models to examine whether downscaling validation had been done for the ecoregions containing our study sites.

While no fisheries ensemble model downscaling had been done specific to our study regions, most of the models of the ensemble have been independently evaluated against separate datasets aggregated at scales down to Large Marine Ecosystems (LMEs) or Exclusive Economic Zones (EEZs) (see[11]). For example, the DBEM was created with the objective of understanding the effects of climate change on exploited marine fish and invertebrate species[2,70]. This model roughly predicts species' habitat suitability; and simulates spatial population dynamics of fish stocks to output biomass and maximum catch potential (MCP), a proxy of maximum sustainable yield[2,62,71]. Compared with spatially-explicit catch data from the Sea Around Us Project (SAUP; www.seaaroundus.org)[70] there were strong similarities in the responses to warming extremes for several EEZs in our current paper (Indonesia and Philippines) and weaker for the EEZs of Madagascar, Papua New Guinea, and Tanzania. At the LME level, DBEM MCP simulations explained about 79% of the variation in the SAUP catch data across LMEs[72]. The four LMEs analyzed in this paper (Agulhas Current; Bay of Bengal; Indonesian Sea; and Sulu-Celebes Sea) fall within the 95% confidence interval of the linear regression relationship[62]. Another example, BOATS, is a dynamic biomass size-spectrum model parameterised to reproduce historical peak catch at the LME scale and observed catch to biomass ratios estimated from the RAM legacy stock assessment database (in 8 LMEs with sufficient data). It explained about 59% of the variability of SAUP peak catch observation at the LME level with the Agulhas Current, Bay of Bengal, and Indonesian Sea catches reproduced within +/-50% of observations[61]. The EcoOcean model validation found that all four LMEs included in this study fit very close to the 1:1 line for overserved and predicted catches in 2000[64,65]. DBPM, FEISTY, and APECOSM have also been independently validated by comparing observed and predicted catches. While the models of this ensemble have used different climate forcings when evaluated independently, when taken together the ensemble multi-model mean reproduces global historical trends in relative biomass, that are consistent with the long term trends and year-on-year variation

in relative biomass change (R$^2$ of 0.96) and maximum yield estimated from stock assessment models (R$^2$ of 0.44) with and without fishing respectively[11].

Crop yield estimates simulated by GGCMI crop models have been evaluated against FAOSTAT national yield statistics[14,73,74]. These studies show that the models, and especially the multi-model mean, capture large parts of the observed inter-annual yield variability across most main producer countries, even though some important management factors that affect observed yield variability (e.g., changes in planting dates, harvest dates, cultivar choices, etc.) are not considered in the models. While GCM-based crop model results are difficult to validate against observations, Jägermeyr et al[14]. show that the CMIP6-based crop model ensemble reproduces the variability of observed yield anomalies much better than CMIP5-based GGCMI simulations. In an earlier crop model ensemble of GGCMI, Müller et al[74]. show that most crop models and the ensemble mean are capable of reproducing the weather-induced yield variability in countries with intensely managed agriculture. In countries where management introduces strong variability to observed data, which cannot be considered by models for lack of management data time series, the weather-induced signal is often low[75], but crop models can reproduce large shares of the weather-induced variability, building trust in their capacity to project climate change impacts[74].

We then attempted to validate the models in our study regions. For the crop models, we examined production-weighted agricultural projections weighted by current yields/production area (Supplementary Fig. 1). We used an observational yield map (SPAM2005) and multiplied it with fractional yield time series simulated by the models to calculate changes in crop production over time, which integrates results in line with observational spatial patterns. The weighted estimates were not significantly different to the unweighted ones (t = 0.17, df = 5, p = 0.87). For the fisheries models, our study regions were data-poor and lacked adequate stock assessment data to extend the observed global agreement of the sensitivity of fish biomass to climate during our reference period (1983-2013). Instead, we provide the degree of model run agreement about the direction of change in the ensemble models to ensure transparency about the uncertainty in this downscaled application.

**Analyses**. To account for the fact that communities were from five different countries we used linear mixed-effects models (with country as a random effect) for all analyses. All averages reported (i.e. exposure, sensitivity, and model agreement) are estimates from these models. In both our comparison of fisheries and agriculture exposure and test of differences between production-weighted and unweighted agriculture exposure we wanted to maintain the paired nature of the data while also accounting for country. To accomplish this we used the differences between the exposure metrics as the response variable (e.g. fisheries exposure minus agriculture exposure), testing whether these differences are different from zero. We also used linear mixed-effects models to quantify relationships between the material style of life and potential impacts under different mitigation scenarios (SSP1-2.6 and 8.5), estimating standard errors from 1000 bootstrap replications. To further explore whether these relationships between the material style of life and potential impacts were driven by exposure or sensitivity, we conducted an additional analysis to quantify relationships between the material style of life and: 1) joint fisheries and agricultural sensitivity; 2) joint fisheries and agricultural exposure under different mitigation scenarios. We present both the conditional $R^2$ (i.e., variance explained by both fixed and random effects) and the marginal $R^2$ (i.e., variance explained by only the fixed effects) to help readers compare among the material style of life relationships.

**Reporting summary**. Further information on research design is available in the Nature Research Reporting Summary linked to this article.

## Data availability
The de-identified exposure, sensitivity, and material style of life data generated in this study for each community can be accessed through Zenodo[76] [https://doi.org/10.5281/zenodo.6496413]. All outputs from the FishMIP model ensemble are available via ISIMIP [https://www.isimip.org/gettingstarted/data-access/]. Raw social survey data are not available because our verbal informed consent made it clear that only aggregated data would be published. The sample sizes and proportions of each community included in the social surveys can be found in the Supplementary Data file. Base layer map data in Fig. 1c and Supplementary Figures 5, 8, and 9 is from Natural Earth, which is freely available through their website (naturalearthdata.com). The SEDAC gridded populating density of the world dataset used to identify a subset of random locations can be found at the following: https://sedac.ciesin.columbia.edu/data/set/gpw-v4-population-density-rev11/data-download.

## Code availability
Code used to analyse and visualize results is available through Zenodo[76] [https://doi.org/10.5281/zenodo.6496413].

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

## Acknowledgements

J.E.C. is supported by the Australian Research Council (CE140100020, FT160100047, DP110101540, and DP0877905). This work was undertaken as part of the Consultative Group for International Agricultural Research (CGIAR) Research Program on Fish Agri-Food Systems (FISH) led by WorldFish. T.D.E acknowledges support from the Natural Sciences and Engineering Research Council of Canada (RGPIN-2021-04319). M.C. and J.S. acknowledge support from the Spanish project ProOceans (RETOS-PID2020-118097RB-I00) and the 'Severo Ochoa Centre of Excellence' accreditation (CEX2019-000928-S) to the Institute of Marine Science (ICM-CSIC). G.G.G. acknowledges support from an Australian Research Council Discovery Early Career Research Award (DE210101918). C.M.P. acknowledges support from NOAA grants NA20OAR4310441 and NA20OAR4310442. M.C. acknowledges the financial support of Ministerio de Ciencia e Innovación, Proyectos de I+D+I (RETOS-PID2020-118097RB-I00, ProOceans) and the institutional support of the 'Severo Ochoa Centre of Excellence' accreditation (CEX2019-000928-S).

## Author contributions

J.E.C. conceived of the study and hosted a workshop with G.G.G., A.D., and R.P. to operationalise the concept. J.E.C., G.G.G., R.P., J.K., N.J., A.R., R.L., A.W., and A.D. provided socioeconomic data. J.J., C.M., C.F., W.L. contributed crop model simulations. J.B., M.C., J.S., T.E., J.E., D.G., J.G., R.F.H., C.N., J.P.A., C.P., and D.T. contributed fisheries model simulations. L.T., J.J., R.F.H., T.E., and I.R.C. analysed the data and all authors contributed to the writing of the manuscript.

## Competing interests

The authors declare no competing interests.

## Additional information

¹ARC Centre of Excellence for Coral Reef Studies, James Cook University, Townsville, QLD 4811, Australia. ²National Center for Scientific Research, PSL Université Paris, CRIOBE, USR 3278, CNRS-EPHE-UPVD, Maison des Océans, 195 rue Saint-Jacques, 75005 Paris, France. ³Moana Ecologic, Rocbaron, France. ⁴Private Fisheries and Environment Consultant, Lau, Morobe, Papua New Guinea. ⁵Institute for Marine and Antarctic Studies, University of Tasmania, Hobart, TAS, Australia. ⁶Center for Marine Socioecology, Hobart, TAS, Australia. ⁷Institute of Marine Science (ICM-CSIC) & Ecopath International Initiative (EII), Barcelona 08003, Spain. ⁸College of Science and Engineering, James Cook University, Building 142, Townsville, QLD 4811, Australia. ⁹Centre for Sustainable Tropical Fisheries and Aquaculture, James Cook University, Townsville, QLD 4811, Australia. ¹⁰Centre for Fisheries Ecosystems Research, Fisheries & Marine Institute, Memorial University of Newfoundland, St. John's, NL, Canada. ¹¹School of Mathematics and Physics, University of Queensland, Brisbane, QLD, Australia. ¹²CSIRO Oceans and Atmosphere, Queensland Biosciences Precinct, St Lucia, QLD, Australia. ¹³Centre for Marine Science and Innovation, School of Biological, Earth and Environmental Sciences, University of New South Wales, Sydney, NSW, Australia. ¹⁴Biodiversity and Natural Resources Program, International Institute for Applied Systems Analysis, Schlossplatz 1, A-2361 Laxenburg, Austria. ¹⁵DECOD (Ecosystem Dynamics and Sustainability), Institut Agro / Inrae / Ifremer, Rennes, France. ¹⁶Department of Atmospheric and Oceanic Sciences, University of California, Los Angeles, CA, USA. ¹⁷School of Mathematical Sciences, Queensland University of Technology, Brisbane, QLD, Australia. ¹⁸NASA Goddard Institute for Space Studies, New York City, NY, USA. ¹⁹Columbia University, Climate School, New York, NY 10025, USA. ²⁰Potsdam Institute for Climate Impact Research (PIK), Member of the Leibniz Association, Potsdam, Germany. ²¹Institute for Marine Science, University of Dar Es Salaam, Zanzibar, Tanzania. ²²Environment and Marine Scientist, New Ireland Province, Papua New Guinea. ²³Wildlife Conservation Society, Goroka, EHP, Papua New Guinea. ²⁴Center for Agricultural Water Research in China, College of Water Resources and Civil Engineering, China Agricultural University, Beijing 100083, China. ²⁵MARBEC, IRD, Univ Montpellier, CNRS, Ifremer, Sète, France. ²⁶Center for Limnology, University of Wisconsin – Madison,

Wisconsin, WI, USA. [27]Institute for the Oceans and Fisheries, The University of British Columbia, Vancouver, BC, Canada. [28]Scripps Institution of Oceanography, University of California, San Diego, CA 92093, USA. [29]Department of Ecology and Evolutionary Biology, University of California, Santa Cruz, Santa Cruz, CA, USA. [30]Department of Biology, Dalhousie University, Halifax, NS B3H 4R2, Canada. [31]United Nations Environment Programme World Conservation Monitoring Centre, 219 Huntingdon Road, Cambridge CB3 0DL, UK. [32]School of Environmental and Earth Sciences, Pwani University, P.O. Box 195 Kilifi, Kenya. [33]Department of Marine Affairs, University of Rhode Island, Kingston, RI 02881, USA. [34]School of Marine & Environmental Affairs, University of Washington, 3707 Brooklyn Avenue NE, Seattle, WA 98105, USA.
✉email: Joshua.cinner@jcu.edu.au

