## [Peer Review File · Nature Communications]

Reviewer comments, first round

Reviewer #1 (Remarks to the Author):

The authors present an exciting study that aims to integrate survey data from coastal communities in 5 countries (that assess material style of life, a metric for material wellbeing, and sensitivity, metrics for dependence on agriculture or fishing, or both) with current and future agricultural and fisheries model output from the Inter-Sectoral Impact Model Intercomparison Project (ISI-MIP) dataset. Their results have important implications in the development of policy and mitigation strategies. The authors find that potential fisheries impacts (exposure) generally larger than those in agriculture, and that most of the 73 sites studied will face important losses in both agriculture and fisheries. Generally, mitigation (a future more like the RCP2.6 scenario as opposed to the RCP8.5 or 4.5 scenario) strongly reduces losses. Since they are more dependent on fisheries and/or agriculture, poorer communities will be subject to higher potential losses.

A strong motivation for this work is that current state-of-the-art intersectoral model output (such as that from the ISI-MIP database) tends to be aggregated over relatively large spatial scales such as countries, and that such scales does not represent the true variability inherent to those regions or the true scale at which socioeconomic impacts take place. This is true and an important ongoing challenge in the community. This study, and the surveys on which the community-level sensitivity observations are based, goes some way to bringing these conceptual frameworks closer together. Such work will eventually allow more refined interpretations of spatial patterns resolved in computational models of fisheries and agriculture, so that local-scale societal impacts can be more appropriately investigated.

The figures are informative and clearly described. The statistical modeling is appropriate and valid given the datasets at hand, and sufficiently documented. As noted in the Reporting Summary, the software/code used to do the analyses are not yet available, and the datasets used are available by request. Despite the developments described in this manuscript, I have two important concerns that prevent me from recommending publication in its current form.

Key comments

The first concerns the RCP scenarios used and is relatively easy to remedy. There is increasing consensus that RCP 8.5 is not a "Business as usual" scenario as much as an unrealistically pessimistic scenario of the future (see Burgess et al., 2020, <https://iopscience.iop.org/article/10.1088/1748-9326/abcdd2>). As your study extends to the period of 2045-2055, the output you use is in the future time frame where RCP 8.5 should no longer be used. At the very least, current thinking on this problematic scenario must be acknowledged. That said, since you have presented output for RCP4.5 (which currently appears to be a better reflection of recent history and coming energy use and technological developments), as well as RCP2.6, I suspect that your arguments for the benefits of mitigation will generally hold, although they will be damped.

More importantly, additional validations will be required to justify the use of the fisheries and agriculture output from ISI-MIP for the 73 sites as described. On lines 55-57, and citing references 14 and 15, the authors note that the simultaneous impacts of climate change on fisheries and agriculture have been examined, but only at national levels. Modelers would happily conduct validations and analyses at higher resolutions, up to and including individual grid points; however, aggregations to nations (for agriculture), and large marine ecosystems (LMEs) or exclusive economic zones (for fisheries) are instead used because current knowledge and model assumptions are: 1) not appropriate at higher-resolution scales and so do not perform well; or 2) there are insufficient or inappropriate data with which to conduct a validation at higher resolution. I am concerned that the authors are using the ISI-MIP agriculture and fisheries output at a scale that it is not necessarily designed for, and a scale that has not been validated.

The authors must be able to show that the ISI-MIP output is appropriate for the scale of their application. I see this taking place in two steps. First, the authors should review how well the ISI-MIP models performed for the regions which contain the study locations. For example, fisheries models were generally validated by Large Marine Ecosystem (LMEs), and so the authors could

confirm how well or poorly the fisheries models represented the LMEs of the study sites. Similarly, then for the agriculture models. This first validation would consist of literature reviews of how well the ISI-MIP models represent these regions.

Second, the authors should show that modeled yield (for agriculture) and the modeled maximum catch potential (for fisheries) over the historical reference period of 2006-2016, is a reasonable reflection of historical observations based on relevant quantities for their study locations. This may be challenging due to differences in what is represented by the models and what data is available. However, a path forward could be to develop and validate a transfer function that relates the modeled output (such as at the scale already used; that is, a radius of 300-350 km around the study locations) to a relevant local agriculture yield or catch. This would not be without problems; however, as one would have to then assume that the transfer function does not change through time to directly apply the model output for the years 2045-2055. It would be ideal if at least one detailed validation could be conducted for each country studied.

Essentially, the first validation would clarify whether the ISI-MIP models used capture the large-scale features of the regions of interest, whereas the second validation would investigate how well the models represent the areas around the study locations.

Minor Comments

Abstract

The methodology presented here is unique, and some sense of the use of community surveys and multi-model intersectoral computational output would be valuable to readers if mentioned in the abstract.

Lines 52-53

Quantifying this would strengthen the article motivation. For example, what fraction of coastal communities are also dependent on agriculture? What can be said about this quantity at the global scale, and in the 5 countries studied?

Figure 1

The circles representing the study site locations are quite large. Mostly this is not problematic, but for the southern and eastern Indonesian sites, it is not evident which coast the location is on.

Line 309

If possible, it would be valuable to know how many grid points are used to calculate each of these averages.

Line 319

As for Line 309.

Reviewer #2 (Remarks to the Author):

The authors aim to address an important gap in the literature by assessing the multi-sectoral implications of climate change in coastal communities. Using household level data from a diversity of communities, this manuscript highlights three key results – where within-country variability is a particularly novel component of each. Firstly, the authors assert that the exposure to loss and sensitivity (“susceptibility to harm from perturbations and adaptive capacity” – line 47) is higher in fisheries than in agriculture. The second and third findings, provide insights into the combined effect on agriculture and fisheries and on wealth strata under three RCP scenarios. These results are used to highlight the importance of global mitigation efforts.

The underlying data appears to be robust and appropriate to support the claims made about the regions studied in the manuscript. Metrics appear appropriate proxies for the phenomena being studied. I would argue that there are risks of biases in the point estimates and standard errors presented in the manuscript (more on this below).

I am not familiar with the literature on fisheries, but the multi-sectoral, sub-national and

geographical coverage appear to be unique aspects of this manuscript. The findings provide useful evidence, where if robust, could highlight the moral imperative of global mitigation efforts. The limitations in generalisability may limit the influence that this manuscript can have on the thinking in the field.

There are a number of methodological components that need to be clarified for the reader. With regards to sampling, I would like to know how were these sites chosen and what these findings are expected to be representative of. Please describe the sample calculation method (e.g. was intra-cluster correlation used). Please tabulate the number sampled, the optimal sample size and indicate where time restricted sampling (in relation to lines 232-233) – for each site in SI and summarise in the manuscript. Also indicate the sampling method for each site (in reference to lines 234 and 235).

The sensitivity analysis is an important step in checking bias towards low or high exposure. It is not clear to me whether grids were randomly selected globally or only in the countries where households were sampled from. If weightings were used in the regressions, then these should also be described.

In the sensitivity sub-section, more information is needed on the ranking of the importance of livelihood activities. There is substantial risk of biases in the responses to these questions, depending on who was asked and at what time of year they were asked (in relation to line 251-252).

The analytical approach will most likely produce biased standard errors. Incorporating some of the hierarchical structure into the regressions is a positive step, but the full structure of the data should be represented in these models – need community within country and consider also nesting within project.

The sensitivity analysis T-test will give biased results in a hierarchical dataset such as this. These data could be analysed using hierarchical (mixed) models.

Overall, I find that the manuscript is well-written and the figures are effective.

Specific suggestions:

- The implications of model assumptions should be incorporated in the summary of limitations (in reference to lines 300-302 & 312)
- The text in figure 3 and figure S1 is too small to be legible. Revise
- Line 119 and 120 – it is not clear that you are talking about global mitigation GHG efforts until reference to RCPs. Rephrase.

Reviewer #3 (Remarks to the Author):

There is a lack of studies that assess the potential impact of climate change on both terrestrial and marine resource systems in coastal communities—this study aims to take “an important first step.” I agree that this is an area where more work needs to be done and I commend the authors for their efforts in approaching this bulky question. The authors do a good job of clearly describing their three main results. I think this manuscript can make an important contribution, especially in building a path for future studies to build upon their ideas. However, I think the authors need to be more clear about the limitations of their approach (of which there are quite a few). I also think that this paper does not include enough of the “so what?” element. I would have liked to see more that highlights why the results of this study are important, how they can be used, etc.

-Lines 49-50: The authors frame the paper by highlighting the need for more localized data on the “key social dimensions of vulnerability.” While they do acknowledge that they are focusing on “potential impacts,” which excludes adaptive capacity, I would argue that they need to, at some point beyond the intro, return to the concept of adaptive capacity and highlight its importance in determining the fate of communities.

-Lines 58-62. This sentence presents the situation as too simplistic. I don't think it is just the "degree" to which people are engaged in these sectors. For example, a household who plants a monoculture of a cash crop and dynamite fishes on the reef may look like they are equally as engaged in both fisheries and agriculture as a household that sustainably farms in an intercropped agroforest and uses a diversity of fishing methods in a diversity of habitats. Perhaps in this sentence, "degree" is the problematic word that needs to be changed. But, I think my point illustrates the limitations of the methods used in this research to determine "sensitivity."

-Lines 64-66: I understand that there is little space to describe the methods in the main text. But, I do think the authors should be clear about the "survey data" used to determine sensitivity in these lines. Essentially, to my understanding, the only data used was the listing and ranking of livelihoods. Yet, in the intro there are several references to "key social dimensions," and this lead up makes it seem like more social data was incorporated into the sensitivity metric.

-Figure 1: need to explain the error bars in the caption

-Lines 127-129: Seems like agricultural losses decrease more than fisheries losses, yet "decrease significantly" is used to describe fisheries in this sentence

-Lines 129-132: These are interesting results for the fisheries sector. Why not present parallel results for agriculture?

-Line 156: careful with the use of "poverty." Your measure of material well-being is not measuring poverty

-Lines 162-169: I appreciate this point, but I think you could condense this argument and provide other potential interventions. Or, suggestions for how these results can be used.

-Line 178: "...are likely to be impacted" This language needs to be changed because you do not consider adaptive capacity, so these are still "potential impacts" and not "likely impacts"

-Line 179: "some weaknesses"—I think there are other limitations (not necessarily weaknesses) that you should discuss in this section. I think there are limitations of the fishery model (e.g., does this model work well across all habitats? Does it capture the complex dynamics of coral reefs where these communities are probably mostly fishing? What about fisheries management—I see that the fisheries model takes into account fishing, but what about potential management practices?)

Another limitation is your sensitivity metric. It is quite simplistic and there are other factors beyond a simple list of livelihoods that determine sensitivity.

Further, crop models for rice, maize, and cassava (Line 297) were used. It looks like you averaged these three crops for every location. Are each of these crops important in each country? Are there other crops that are more important in some of these countries (e.g., sweet potato, yams, taro) for which data was not available or used?

-Line 270: Do you use this measure of both sector sensitivity anywhere?

Reviewer #1 (Remarks to the Author):

As noted in the Reporting Summary, the software/code used to do the analyses are not yet available, and the datasets used are available by request.

These are now included

...The first concerns the RCP scenarios used and is relatively easy to remedy. There is increasing consensus that RCP 8.5 is not a “Business as usual” scenario as much as an unrealistically pessimistic scenario of the future (see Burgess et al., 2020, <https://iopscience.iop.org/article/10.1088/1748-9326/abcd2>). As your study extends to the period of 2045-2055, the output you use is in the future time frame where RCP 8.5 should no longer be used. At the very least, current thinking on this problematic scenario must be acknowledged. That said, since you have presented output for RCP4.5 (which currently appears to be a better reflection of recent history and coming energy use and technological developments), as well as RCP2.6, I suspect that your arguments for the benefits of mitigation will generally hold, although they will be damped.

As suggested by the editor, we have removed the terminology “business as usual” but kept the scenario. We have added the following note about critical thinking on this scenario “...Specifically, exposure under the high-emissions Shared Socioeconomic Pathway 8.5 scenario (which has tracked historic cumulative CO² emissions¹⁷, but has been recently critiqued for over-projecting CO² emissions and economic growth¹⁸)...” and made a specific note about this in our “critiques and caveats” section- “Lastly, we used shared socioeconomic pathway scenarios that bracket the full range of scenario variability (SSP5-8.5 and SSP1-2.6). At the time of publication, these were the only scenarios available for both fisheries and agriculture using the Fastrack Phase 3 dataset. Future publications may wish to explore additional scenarios.”

More importantly, additional validations will be required to justify the use of the fisheries and agriculture output from ISI-MIP for the 73 sites as described. On lines 55-57, and citing references 14 and 15, the authors note that the simultaneous impacts of climate change on fisheries and agriculture have been examined, but only at national levels. Modelers would happily conduct validations and analyses at higher resolutions, up to and including individual grid points; however, aggregations to nations (for agriculture), and large marine ecosystems (LMEs) or exclusive economic zones (for fisheries) are instead used because current knowledge and model assumptions are: 1) not appropriate at higher-resolution scales and so do not perform well; or 2) there are

insufficient or inappropriate data with which to conduct a validation at higher resolution. I am concerned that the authors are using the ISI-MIP agriculture and fisheries output at a scale that it is not necessarily designed for, and a scale that has not been validated.

The authors must be able to show that the ISI-MIP output is appropriate for the scale of their application. I see this taking place in two steps. First, the authors should review how well the ISI-MIP models performed for the regions which contain the study locations. For example, fisheries models were generally validated by Large Marine Ecosystem (LMEs), and so the authors could confirm how well or poorly the fisheries models represented the LMEs of the study sites. Similarly, then for the agriculture models. This first validation would consist of literature reviews of how well the ISI-MIP models represent these regions.

Second, the authors should show that modeled yield (for agriculture) and the modeled maximum catch potential (for fisheries) over the historical reference period of 2006-2016, is a reasonable reflection of historical observations based on relevant quantities for their study locations. This may be challenging due to differences in what is represented by the models and what data is available. However, a path forward could be to develop and validate a transfer function that relates the modeled output (such as at the scale already used; that is, a radius of 300-350 km around the study locations) to a relevant local agriculture yield or catch. This would not be without problems; however, as one would have to then assume that the transfer function does not change through time to directly apply the model output for the years 2045-2055. It would be ideal if at least one detailed validation could be conducted for each country studied.

Essentially, the first validation would clarify whether the ISI-MIP models used capture the large-scale features of the regions of interest, whereas the second validation would investigate how well the models represent the areas around the study locations.

We put a lot of effort into addressing this important comment, and it led to a collaboration with the model developers from both the Fish-MIP and Ag-MIP. This led to two substantive changes and some additions. First, we used the recently released CMIP6-based simulations instead of the former fast track data (see for example, Tittensor, D. P. *et al.* Next-generation ensemble projections reveal higher climate risks for marine ecosystems. *Nature Climate Change* 1–9 (2021)). Second, based on advice from the AgMIP coordinator, we shifted to the transient CO2 results for agricultural production, which is now the community standard (instead of the constant CO2 we had previously used). The additions entailed making transparent the degree of model agreement, and a sensitivity test exploring how using different numbers of cells changes the model agreement.

To address the reviewers first point about validation, we conferred with colleagues at the Fish-MIP project, who have published a series of papers which attempts to address these issues for other ecoregions. We now added a paragraph stating “We attempted a two-stage validation of the ensemble model projections. First, we

reviewed the literature on downscaling of ensemble models to examine whether downscaling validation had been done for the ecoregions containing our study sites.

While no fisheries ensemble model downscaling had been done specific to our study regions, most of the models of the ensemble have been independently evaluated against separate datasets aggregated at scales down to Large Marine Ecosystems (LMEs) or Exclusive Economic Zones (EEZs) (see ¹¹). For example, the DBEM was created with the objective of understanding the effects of climate change on exploited marine fish and invertebrate species^{2,71}. This model roughly predicts species' habitat suitability; and simulates spatial population dynamics of fish stocks to output biomass and maximum catch potential (MCP), a proxy of maximum sustainable yield^{2,62,70}. Compared with spatially explicit catch data from the Sea Around Us Project (SAUP; www.seaaroundus.org)⁷¹ there were strong similarities in the responses to warming extremes for several EEZs in our current paper (Indonesia and Philippines) and weaker for the EEZs of Madagascar, Papua New Guinea, and Tanzania. At the LME level, DBEM MCP simulations explained about 79% of the variation in the SAUP catch data across LMEs⁷². The four LMEs analyzed in this paper (Agulhas Current; Bay of Bengal; Indonesian Sea; and Sulu-Celebes Sea) fall within the 95% confidence interval of the linear regression relationship⁶². Another example, BOATS, is a dynamic biomass size-spectrum model parameterised to reproduce historical peak catch at the LME scale and observed catch to biomass ratios estimated from the RAM legacy stock assessment database (in 8 LMEs with sufficient data). It explained about 59% of the variability of SAUP peak catch observation at the LME level with the Agulhas Current, Bay of Bengal, and Indonesian Sea catches reproduced within +/-50% of observations⁶¹. The EcoOcean model validation found that all four LMEs included in this study fit very close to the 1:1 line for overserved and predicted catches in 2000^{64,65}. DBPM, FEISTY, and APECOSM have also been independently validated by comparing observed and predicted catches. While the models of this ensemble have used different climate forcings when evaluated independently, when taken together the ensemble multi-model mean reproduces global historical trends in relative biomass, that are consistent with the long term trends and year-on-year variation in relative biomass change (R^2 of 0.96) and maximum yield estimated from stock assessment models (R^2 of 0.44) with and without fishing respectively¹¹.

Crop yield estimates simulated by GGCM crop models have been evaluated against FAOSTAT national yield statistics^{14,73,74}. These studies show that the models, and especially the multi-model mean, capture large parts of the observed inter-annual yield variability across most main producer countries, even though some important management factors that affect observed yield variability (e.g., changes in planting dates, harvest dates, cultivar choices, etc.) are not considered in the models. While GCM-based crop model results are difficult to validate against observations, Jägermeyr et al.¹⁴ show that the CMIP6-based crop model ensemble reproduces the variability of observed yield anomalies much better than CMIP5-based GGCM simulations. In an earlier crop model ensemble of GGCM, Müller et al.⁷⁴ show that most crop models and the ensemble mean are capable of reproducing the weather-induced yield variability in countries with intensely managed agriculture. In countries where management introduces strong variability to observed data, which cannot be considered by models for lack of management data time series, the weather-induced signal is often low⁷⁵, but crop models can reproduce large shares of the weather-induced variability, building trust in their capacity to project climate change impacts⁷⁴.

To address the reviewers second point about validation against observations, we attempted to estimate the realism of historical fish biomass variation around our study regions, but unfortunately our study regions are too data poor for comparison. We looked through EcoBase (a database of all Ecopath with Ecosim models) but few models from our study countries were completed well before our timeline of interest (generally 1970s and a few in the 1990s). Instead, we focused on the level of agreement of the ensemble of model simulations. Models used for the ISIMIP projections for fisheries have been independently developed, evaluated, and together reproduce observed global biomass decrease as previously mentioned (see Lotze et al. 2019). Our working assumption is that areas around study locations associated with high agreement between models are associated with more confidence. For the Agricultural data, we also conducted a supplemental yield-based bias correction. We use an observational yield map (SPAM2005) and multiply it with fractional yield time series simulated by the models to calculate changes in crop production over time. This way, by definition, the results are in line with observational spatial patterns. These are now presented as a supplemental figure and are compared with the unweighted model outputs (there is no difference).

We added a paragraph which reads “We then attempted to validate the models in our study regions. For the crop models, we examined production-weighted agricultural projections weighted by current yields/production area (Fig. S1). We used an observational yield map (SPAM2005) and multiplied it with fractional yield time series simulated by the models to calculate changes in crop production over time, which integrates results in line with observational spatial patterns. The weighted estimates were not significantly different to the unweighted ones ($t=0.17$, $df=5$, $p=0.87$). For the fisheries models, our study regions were data poor and lacked adequate stock assessment data to extend the observed global agreement of the sensitivity of fish biomass to climate during our reference period (1983-2013). Instead, we provide the degree of model run agreement about the direction of change in the ensemble models to ensure transparency about the uncertainty in this downscaled application.”

Minor Comments

Abstract

The methodology presented here is unique, and some sense of the use of community surveys and multi-model intersectoral computational output would be valuable to readers if mentioned in the abstract

Done. The abstract now includes the following: “Here, we combine socioeconomic surveys and multi-model intersectoral computational outputs to conduct a sub-national analysis of the potential impacts of climate change on fisheries and agriculture in 72 coastal communities across 5 countries.”

Lines 52-53

Quantifying this would strengthen the article motivation. For example, what fraction of

coastal communities are also dependent on agriculture? What can be said about this quantity at the global scale, and in the 5 countries studied?

We were unable to fulfil this request at this specific point in the introduction, but do add a paragraph about it using our data further down. To my knowledge this has never been quantified, either at the national or global scale for coastal communities. I have written a lot about coastal livelihoods (see for example Cinner, J.E. (2014). Coral reef livelihoods. *Current Opinion in Environmental Sustainability*. 7:65-71 and Cinner, J.E. and Ö. Bodin. (2010). Livelihood diversification in tropical coastal communities: a network-based approach to analyzing 'livelihood landscapes'. *PLoS One*. 5:e11999.), but have never analysed specifically what you are suggesting, nor seen it analysed. Thus, we now write:

“Over a third of our sites (36% under SSP5-8.5) are expected to experience potential increases in agriculture (due to CO² fertilization effects that fuel potential increases particularly in rice yields) while experiencing potential losses in fisheries. For these sites, a question of critical concern is whether the potential gains in agriculture could help offset the losses in fisheries. The answer to this lies in part on the degree of substitutability between sectors. Our survey of 3008 households reveals high variation among countries, and even within some countries in the degree of household occupational multiplicity incorporating both agriculture and fisheries sectors (Table 1). 31% of households in our study engaged in both fishing and agriculture, though this ranged from 10% of households in the Philippines to 77% of households in Papua New Guinea. This means that the degree to which agricultural gains might possibly offset some fisheries losses at the household scale is very context dependent. Our survey also revealed that 17% of households were involved in agriculture but not fisheries, ranging from 33% in Madagascar to 3% in our Papua New Guinean study communities. Alternatively, more than a third of households surveyed in Indonesia and Philippines were involved in fisheries but not agriculture (36% and 37% respectively), compared to a low of 16% in Madagascar. In 12% of the Philippines communities surveyed (n=3), not a single household was engaged on agriculture. Thus, for 32% of households across our sample, including some entire communities, potential agricultural gains will not offset potential fisheries losses. In these locations building adaptive capacity to buffer change will be critical²¹.”

Figure 1. The circles representing the study site locations are quite large. Mostly this is not problematic, but for the southern and eastern Indonesian sites, it is not evident which coast the location is on.

We have added an inset map in the SI. We tried having a map with smaller point sizes, but we now also show the model agreement, and this could not be seen with smaller point sizes..

Line 309

If possible, it would be valuable to know how many grid points are used to calculate each of these averages.

This is now specified in the methods and visually demonstrated in Fig S5 for both agriculture and fisheries.

Line 319

As for Line 309.

Reviewer #2 (Remarks to the Author):

There are a number of methodological components that need to be clarified for the reader. With regards to sampling, I would like to know how were these sites chosen and what these findings are expected to be representative of.

To address this comment, we have added the following “Within each project, sites were purposively selected to be representative of the broad range of socioeconomic conditions (e.g., population size, levels of development, integration to markets) experienced within the region.”

Please describe the sample calculation method (e.g. was intra-cluster correlation used).

The sample calculation method varied by project (this paper synthesised data from 5 different projects- each used a similar method of surveying, but different sampling approach). This is now made explicit in Table S1.

Please tabulate the number sampled, the optimal sample size and indicate where time restricted sampling (in relation to lines 232-233) – for each site in SI and summarise in the manuscript. Also indicate the sampling method for each site (in reference to lines 234 and 235).

Done. This is a new supplemental table.

The sensitivity analysis is an important step in checking bias towards low or high exposure. It is not clear to me whether grids were randomly selected globally or only in the countries where households were sampled from.

To address this comment, we have added the following “...of the 5 countries we studied...” to the sensitivity test section of the methods

If weightings were used the the regressions, then these should also be described.

Linear mixed effects models were used for all analyses so that we could account for the effect of country in all estimates, comparisons, and assessment of relationships. This is now more fully described in the methods section, where we now write “To account for the fact that communities were from five different countries we used linear mixed effects models (with country as a random effect) for all analyses. All averages reported (i.e. exposure, sensitivity, and model agreement) are estimates from these models. In both our

comparison of fisheries and agriculture exposure and test of differences between production-weighted and unweighted agriculture exposure we wanted to maintain the paired nature of the data while also accounting for country; To accomplish this we used the differences between the exposure metrics as the response variable (e.g. fisheries exposure minus agriculture exposure), testing whether these differences are different from zero. We also used linear mixed effects models to quantify relationships between material style of life and potential impacts under different mitigation scenarios (SSP1-2.6 and 8.5), estimating 95% confidence intervals from 1000 bootstrap replications. To further explore whether these relationships between material style of life and potential impacts were driven by exposure or sensitivity, we conducted a supplemental analysis to quantify relationships between material style of life and: 1) joint fisheries and agricultural sensitivity; 2) joint fisheries and agricultural exposure under different mitigation scenarios. We present both the conditional R^2 (i.e., variance explained by both fixed and random effects) and the marginal R^2 (i.e., variance explained by only the fixed effects) to help readers compare among the material style of life relationships.”

In the sensitivity sub-section, more information is needed on the ranking of the importance of livelihood activities. There is substantial risk of biases in the responses to these questions, depending on who was asked and at what time of year they were asked (in relation to line 251-252).

To address this comment, we have now added the following information for clarity

“Respondents were generally the household head, but could have been other household members if the household head was not available during the study period (i.e. was away).”

And in regards to our sensitivity test, we now note:

These surveys were semi-panel data (i.e. the community was surveyed repeatedly, but we did not track individuals over each sampling interval) and sometimes occurred in different seasons.

And

“Although our survey methodology has the potential for bias (e.g. people might provide different rankings based on the season, or there might be gendered differences in how people rank the importance of different occupations³⁵), our time-series analysis suggest that seasonal and potential respondent variation do not dramatically alter our community-scale sensitivity metric.”

The analytical approach will most likely produce biased standard errors. Incorporating some of the hierarchical structure into the regressions is a positive step, but the full

structure of the data should be represented in these models – need community within country and consider also nesting within project.

Yes, the full structure is represented as best it can be. The unit of analysis is community, which is nested within country. It was not possible to nest within project (in some cases projects spanned multiple countries and in others there were multiple projects per country).

The sensitivity analysis T-test will give biased results in a hierarchical dataset such as this. These data could be analysed using hierarchical (mixed) models.

To address this comment, we have replaced the T-Test with a hierarchical model (linear mixed effects model) which is not more fully described in the paragraph on “Analyses”.

Overall, I find that the manuscript is well-written and the figures are effective.

Thanks

Specific suggestions:

- The implications of model assumptions should be incorporated in the summary of limitations (in reference to lines 300-302 & 312)

To address this comment, we have included the following paragraph “

Second there are key limitations and assumptions to the models we used. For example, many tropical small-scale fisheries target seagrass³² and coral reefs³³, the dynamics of which may not be well captured in the global ensemble models. Additionally, the ensemble models were developed at lower resolution and validations at higher resolutions are still lacking. Our approach for dealing with this was to make transparent the degree of ensemble model agreement about the direction of change, which relies on the assumption that we have greater confidence in projections that have higher model agreement. Another limitation is that there may be discrepancies between what is projected and what is actually harvested. For example, many of the fisheries models project changes to non-target species. Likewise, we included just 3 crops in the agricultural models (rice, maize, and cassava), which are key in the study region, with many study countries growing 2 or more of these crops. For example, Indonesia is the 3rd largest producer of rice in the world, and the 6th largest producer of maize and cassava³⁴. However, subsistence agriculture in Papua New Guinea is dominated by banana and yams, for which agricultural yield projections were not available. We used an unweighted average of projected changes in these three crops to represent a portfolio of small-scale agriculture, with a sensitivity test based on agricultural projections weighted by current yields/production area proportions of current yields (Fig. S1). Finally, it is important to keep key model assumptions in mind when interpreting these data, including that the agricultural models assumed no changes in farm management or climate change adaptation over time, while the fisheries models assumed no diazotrophs.

“

- The text in figure 3 and figure S1 is too small to be legible. Revise

We have increased the font size in these figures

- Line 119 and 120 – it is not clear that you are talking about global mitigation GHG efforts until reference to RCPs. Rephrase.

Rephrased to now read “but mitigation efforts that reduce greenhouse gas emissions could dramatically change that”

Reviewer #3 (Remarks to the Author):

I think this manuscript can make an important contribution, especially in building a path for future studies to build upon their ideas. However, I think the authors need to be more clear about the limitations of their approach (of which there are quite a few).

This concern is also captured by Reviewer #2, to which we have substantially expanded the critiques and caveats section (as detailed above and below).

I also think that this paper does not include enough of the “so what?” element. I would have liked to see more that highlights why the results of this study are important, how they can be used, etc.

To address this comment, we have added the following:

“To put these losses in perspective, Sala et al.¹⁸ found that strategically protecting 28% of the ocean could increase food provisioning by 5.9 million tonnes, which is just 6.9% of the 84.4 million tons of marine capture globally in 2018¹⁹. Thus, the mean fisheries losses are approximately double that which could be buffered by strategic conservation.”

And

“

Over a third of our sites (36% under SSP5-8.5) are expected to experience increases in agriculture (due to CO₂ fertilization effects that fuel potential increases particularly in rice yields) while experiencing losses in fisheries catch potential. For these sites, a question of critical concern is whether the potential gains in agriculture could help offset the losses in fisheries catch potential. The answer to this lies in part in the degree of substitutability between sectors. Our survey of 3008 households reveals high variation among countries, and even within some countries in the degree of household occupational multiplicity incorporating both agriculture and fisheries sectors (Table 1). 31% of households in our study

engaged in both fishing and agriculture, though this ranged from 10% of households in the Philippines to 77% of households in Papua New Guinea. This means that the degree to which agricultural gains might possibly offset some fisheries losses at the household scale is very context dependent. Our survey also revealed that 17% of households were involved in agriculture but not fisheries, ranging from 33% in Madagascar to 3% in our Papua New Guinean study communities. Alternatively, more than a third of households surveyed in Indonesia and Philippines were involved in fisheries but not agriculture (36% and 37% respectively), compared to a low value of 16% in Madagascar. In 12% of the Philippines communities surveyed (n=3), not a single household was engaged in agriculture. Thus, for 32% of households across our sample, including some entire communities, potential agricultural gains will not offset potential fisheries losses. In these locations building adaptive capacity to buffer change will be critical⁹.

-Lines 49-50: The authors frame the paper by highlighting the need for more localized data on the “key social dimensions of vulnerability.” While they do acknowledge that they are focusing on “potential impacts,” which excludes adaptive capacity, I would argue that they need to, at some point beyond the intro, return to the concept of adaptive capacity and highlight its importance in determining the fate of communities.

To address this comment, we have added the following “Fourth, our study explicitly focused on the potential impacts of climate change in 72 Indo-Pacific coastal communities by examining their sensitivity and exposure, but our methodology did not enable us to incorporate adaptive capacity. Adaptive capacity is a latent trait that enables people to adapt to and take advantage of the opportunities created by change^{39,40}, and is critically important in determining the fate of coastal communities under climate change. Adaptive capacity is thought to consist of dimensions of assets, flexibility, social organisation, learning, socio-cognitive, and agency^{9,41,42}. Unfortunately, indicators of these dimensions of adaptive capacity were not collected in a standardised manner across all of the different projects comprising this study.

“

-Lines 58-62. This sentence presents the situation as too simplistic. I don't think it is just the “degree” to which people are engaged in these sectors. For example, a household who plants a monoculture of a cash crop and dynamite fishes on the reef may look like they are equally as engaged in both fisheries and agriculture as a household that sustainably farms in an intercropped agroforest and uses a diversity of fishing methods in a diversity of habitats. Perhaps in this sentence, “degree” is the problematic word that needs to be changed. But, I think my point illustrates the limitations of the methods used in this research to determine “sensitivity.”

To address this comment, we have changed the sentences to remove degree. They now read “but this coarse scale does not capture whether people simultaneously engage with- and are likely to be affected by- changes in these sectors. Indeed, whether households engage in both fisheries and agriculture⁹ will determine whether people have the knowledge, skills, and capital to substitute sectors if one declines, or alternatively, make them particularly susceptible to the potential ‘perfect storm’ of a combined decline across sectors¹⁴.”

-Lines 64-66: I understand that there is little space to describe the methods in the main text. But, I do think the authors should be clear about the “survey data” used to determine sensitivity in these lines. Essentially, to my understanding, the only data used was the listing and ranking of livelihoods. Yet, in the intro there are several references to “key social dimensions,” and this lead up makes it seem like more social data was incorporated into the sensitivity metric.

To address this comment, we have added more details about the specific type of data. Specifically, we state “Here, we combine a measure of exposure based on model projections of losses to exploitable marine biomass (here dubbed “fisheries catch potential”) and agriculture from the Inter-Sectoral Impact Model Intercomparison Project (ISIMIP) Fast Track phase 3 dataset with a measure of sensitivity based on survey data about material wealth and engagement in fisheries, agriculture, and other occupational sectors from >3,000 households across 72 tropical coastal communities in five countries (Table S1).” ...

-Figure 1: need to explain the error bars in the caption

Done. Caption now reads “Error bars show 25 and 75% percentiles of exposure.”

-Lines 127-129: Seems like agricultural losses decrease more than fisheries losses, yet “decrease significantly” is used to describe fisheries in this sentence

These results have changed as a result of altering the assumptions used in the agricultural models and by using the most updated projections. Agriculture is no longer expected to experience a significant change from 0 (reference period).

-Lines 129-132: These are interesting results for the fisheries sector. Why not present parallel results for agriculture?

This section now reads “The second key result from our integrated assessment reveals that some locations will bear a double burden of losses to fisheries and agriculture simultaneously, but mitigation efforts that reduce greenhouse gas emissions could curb these losses. Specifically, under SSP5-8.5, 64% of our study sites are expected to lose productivity in fisheries and agriculture simultaneously (Fig. 3A), but this would reduce to 37% of sites under the low emissions scenario SSP1-2.6 (Fig. 3B). Again, the effect of mitigation is consistent in the random selection of 4,746 sites (Figure S2), with 70% of

randomly selected sites expected to experience a double burden under SSP5 8.5, and 47% under SSP1 2.6. Many of the sites expected to experience the highest losses to both fisheries catch potential and agriculture have moderate to high sensitivity (Fig 3A), which means the impacts of these changes could be profoundly felt by coastal communities."

-Line 156: careful with the use of "poverty." Your measure of material well-being is not measuring poverty

To address this comment, we have modified the terminology. At the suggestion of the editor we have used socioeconomic status.

-Lines 162-169: I appreciate this point, but I think you could condense this argument and provide other potential interventions. Or, suggestions for how these results can be used.

To address this comment, we have expanded this section to read :One potential interpretation of our findings is that alternative livelihood programs (e.g. jobs outside the fisheries or agricultural sectors, such as the service industry) could reduce sensitivity in lower socioeconomic status communities. However, decades of research on livelihood diversification has highlighted a multitude of reasons why alternative livelihood projects frequently fail²⁶, including that they do not provide high levels of non-economic satisfactions (e.g., social, psychological, and cultural)²⁷⁻²⁹, as well as cultural barriers to switching occupations (e.g. caste systems)³⁰, and attachment to identity and place³¹. Alternative occupations need to provide some of the same satisfactions, including basic needs (safety, income), social and psychological needs (time away from home, community in which you live, etc.), and self-actualization (adventure, challenge, opportunity to be own boss, etc.). For example, fishing attracts individuals manifesting a personality configuration referred to as an externalizing disposition, which is characterized by a need for challenges, adventure, and risk. Fishing can be extremely satisfying for people with this personality complex, while many alternative occupations can lead to job dissatisfaction, which has negative social and psychological consequences^{32,33}. Research has shown that for fisheries, recreational fishing captains or guides as alternative occupations produce some of the same satisfactions and have been successful³³. Despite these limited successes, alternative livelihood programs frequently fail and are not a viable substitute for mitigating climate change for the ~6 million coral reef fishers globally³⁴.

-Line 178: "...are likely to be impacted" This language needs to be changed because you do not consider adaptive capacity, so these are still "potential impacts" and not "likely impacts"

We have modified this sentence to read “Our study was an important first step in examining the potential simultaneous impacts to fisheries and agriculture in coastal communities ”

-Line 179: “some weaknesses”—I think there are other limitations (not necessarily weaknesses) that you should discuss in this section. I think there are limitations of the fishery model (e.g., does this model work well across all habitats? Does it capture the complex dynamics of coral reefs where these communities are probably mostly fishing? What about fisheries management—I see that the fisheries model takes into account fishing, but what about potential management practices?)

To address this comment, we have added the following “there are key limitations and assumptions to the models we used. For example, many tropical small-scale fisheries target seagrass³⁵ and coral reef habitats³⁴, which are not represented in the global ensemble models. “

We did not include mention of fisheries management in the limitations section, because Lotze et al. studied the models with and without fishing and found a similar magnitude and variability of the climate change effect. Thus, the climate change effect would still be similar even if the fisheries were fully protected from fishing. However, as noted above, we did bring it up earlier in the manuscript. We have now included “To put these losses in perspective, Sala et al.¹⁸ found that strategically protecting 28% of the ocean could increase food provisioning by 5.9 million tonnes, which is just 6.9% of the 84.4 million tons of marine capture globally in 2018¹⁹. Thus, the mean fisheries losses are approximately double that which could be buffered by optimal conservation.”

Another limitation is your sensitivity metric. It is quite simplistic and there are other factors beyond a simple list of livelihoods that determine sensitivity.

To address this comment, we have added the following “Third, our sensitivity metric examined a somewhat narrow aspect of what makes people sensitive to climate change. Sensitivity is thought to contain dimensions of economic, demographic, psychological, and cultural dependency³³. Our metric was based on people’s engagement in natural resource-based livelihoods, which primarily captures the economic dimensions (although livelihoods do provide cultural and psychological contributions to people^{24,26,27,29,34}). “

Further, crop models for rice, maize, and cassava (Line 297) were used. It looks like you averaged these three crops for every location. Are each of these crops important in each country? Are there other crops that are more important in some of these countries (e.g., sweet potato, yams, taro) for which data was not available or used?

To address this comment, we have added the following “we included just 3 crops in the agricultural models (rice, maize, and cassava), which are key in the study region, with many study countries growing 2 or more of these crops. For example, Indonesia is the 3rd largest producer of rice in the world, and the 6th largest producer of maize and cassava³⁶. However, subsistence agriculture in Papua New Guinea is dominated by banana and yams, for which agricultural yield projections were not available. We used an unweighted average of projected changes in these three crops to represent a portfolio of small-scale agriculture, with a sensitivity test based on agricultural projections weighted by current yields/production area proportions of current yields (Fig. S1).”

-Line 270: Do you use this measure of both sector sensitivity anywhere?

Yes, it is the bubble size in figure 3a.

Reviewer comments, second round

Reviewer #1 (Remarks to the Author):

Major Comments

The following review is of the second version of the Cinner et al. manuscript entitled “The potential impacts of climate change on agriculture and fisheries production in 72 tropical coastal communities”.

I recommended three improvements in my review of the first version of this manuscript: 1) Change the use of the RCP scenarios and the (previously) prevailing use of the term “business as usual” for the RCP 8.5 scenario, 2) validate the fisheries and agricultural impact models in the regions pertinent to the study, and 3) validate the historical fisheries and agricultural yields near to the sites considered in the study.

Regarding the first recommendation, the authors adapted a newer version of the fisheries and agricultural impact output from ISI-MIP and refined their interpretation of the scenarios. They now instead use the SSP1-2.6 and SSP5-8.5 scenarios to bound possible future change and gauge the value of mitigation.

For recommendation 2, the authors conferred with the ISI-MIP model developers and carefully detailed previous model validations in the EEZs pertinent to the study, including a comparison to historic fisheries catch data and crop yields.

For agriculture, the authors validated the sites considered in the study (recommendation 3) by means of a bias correction of simulated ISI-MIP agricultural yields. Although the authors were not able to obtain direct observations or other more specialized modeling for fisheries at their study sites, it is acceptable to use the consistency among the ISI-MIP models for this purpose, given that the large-scale ecosystems have been validated (recommendation 2). The limitations of this approach are clearly stated.

Minor Comments

Merge first and second paragraphs

Reviewer #2 (Remarks to the Author):

Thank you for the detailed responses to the points raised in my initial review. This will be a positive contribution to the literature.

Reviewer #3 (Remarks to the Author):

Thank you for the opportunity to review the revised manuscript. I found the edits and responses to the reviewers' comments to be satisfactory and believe the manuscript is now in a form to make an important contribution to the peer reviewed literature. I commend the authors for their efforts!

One minor comment:

I had to read Lines 269-271 a couple of times to comprehend. I suggest editing to something like:

Research has shown that recreational fishing captain or guide jobs produce some of the same satisfactions as fishing and have been successfully introduced as alternative occupations.

RESPONSES TO REVIEWERS COMMENTS

REVIEWERS' COMMENTS

Reviewer #1 (Remarks to the Author):

Major Comments

The following review is of the second version of the Cinner et al. manuscript entitled “The potential impacts of climate change on agriculture and fisheries production in 72 tropical coastal communities”.

I recommended three improvements in my review of the first version of this manuscript: 1) Change the use of the RCP scenarios and the (previously) prevailing use of the term “business as usual” for the RCP 8.5 scenario, 2) validate the fisheries and agricultural impact models in the regions pertinent to the study, and 3) validate the historical fisheries and agricultural yields near to the sites considered in the study.

Regarding the first recommendation, the authors adapted a newer version of the fisheries and agricultural impact output from ISI-MIP and refined their interpretation of the scenarios. They now instead use the SSP1-2.6 and SSP5-8.5 scenarios to bound possible future change and gauge the value of mitigation.

For recommendation 2, the authors conferred with the ISI-MIP model developers and carefully detailed previous model validations in the EEZs pertinent to the study, including a comparison to historic fisheries catch data and crop yields.

For agriculture, the authors validated the sites considered in the study (recommendation 3) by means of a bias correction of simulated ISI-MIP agricultural yields. Although the authors were not able to obtain direct observations or other more specialized modeling for fisheries at their study sites, it is acceptable to use the consistency among the ISI-MIP models for this purpose, given that the large-scale ecosystems have been validated (recommendation 2). The limitations of this approach are clearly stated.

Minor Comments

Merge first and second paragraphs

RESPONSE: First and second paragraphs were merged as suggested

Reviewer #2 (Remarks to the Author):

Thank you for the detailed responses to the points raised in my initial review. This will be a positive contribution to the literature.

Reviewer #3 (Remarks to the Author):

Thank you for the opportunity to review the revised manuscript. I found the edits and responses to the reviewers' comments to be satisfactory and believe the manuscript is now in

a form to make an important contribution to the peer reviewed literature. I commend the authors for their efforts!

One minor comment:

I had to read Lines 269-271 a couple of times to comprehend. I suggest editing to something like:

Research has shown that recreational fishing captain or guide jobs produce some of the same satisfactions as fishing and have been successfully introduced as alternative occupations.

RESPONSE: Text edited as suggested